# Assessing the Impact of AR HUDs and Risk Level on User Experience in Self-Driving Cars: Results from a Realistic Driving Simulation

**Seungju Kim** [1,†], **Jungseok Oh** [1,†], **Minwoo Seong** [1,†], **Eunki Jeon** [1], **Yeon-Kug Moon** [2] **and Seungjun Kim** [1,*]

1   Gwangju Institute of Science and Technology, School of Integrated Technology,
    Gwangju 61005, Republic of Korea
2   Korea Electronics Technology Institute, Seongnam-si 13509, Republic of Korea
*   Correspondence: seungjun@gist.ac.kr
†   These authors contributed equally to this work.

**Abstract:** The adoption of self-driving technologies requires addressing public concerns about their reliability and trustworthiness. To understand how user experience in self-driving vehicles is influenced by the level of risk and head-up display (HUD) information, using virtual reality (VR) and a motion simulator, we simulated risky situations including accidents with HUD information provided under different conditions. The findings revealed how HUD information related to the immediate environment and the accident's severity influenced the user experience (UX). Further, we investigated galvanic skin response (GSR) and self-reported emotion (Valence and Arousal) annotation data and analyzed correlations between them. The results indicate significant differences and correlations between GSR data and self-reported annotation data depending on the level of risk and whether or not information was provisioned through HUD. Hence, VR simulations combined with motion platforms can be used to observe the UX (trust, perceived safety, situation awareness, immersion and presence, and reaction to events) of self-driving vehicles while controlling the road conditions such as risky situations. Our results indicate that HUD information provision significantly increases trust and situation awareness of the users, thus improving the user experience in self-driving vehicles.

**Keywords:** autonomous vehicles; driving simulator; human-machine interaction; physiological signal; reliability; user experience; virtual reality

## 1. Introduction

In recent years, the development of self-driving vehicles has received significant attention because they have the potential to navigate the roads without human intervention. Self-driving vehicles offer better driving experiences with increased safety [1] and traffic efficiency [2]. However, despite the comfort that self-driving vehicles are expected to provide, consumers are known not to fully trust self-driving cars [3]. For the market to adopt self-driving technology, drivers must trust that it is safe; thus, knowing the factors that affect drivers' trust is crucial.

Additionally, the transparency of the artificial intelligence (AI) model employed is considered an essential requirement for self-driving vehicles because it significantly affects drivers' trust [4]. AI models are widely accepted in self-driving areas because of their high performance and decision-making efficiency. To reflect transparency features in AI models, research has been conducted into explainable AIs (XAI) to apprise drivers of the results of the AI's decision-making [5,6]. One method for visualizing the decisions made by the AI in the autonomous vehicle is to use head-up displays (HUDs) [7]. These are displays that appear in the driver's forward-looking line of sight while driving, thus, providing information through a HUD can effectively build the drivers' trust [8].

In this study, we chose the level of risk and that of the road information provision on HUDs as two dimensions, based on previous studies [9,10]. Morra et al. showed that providing evidence that supports self-driving vehicles' decisions is important for cultivating trust [10]. Ajenaghughrure et al. observed changes in trust according to the degree of risk. Although these factors affect trust in self-driving vehicles, only few studies have considered both factors in a single scenario. Based on these insights, we developed realistic driving content in virtual reality (VR) that simulates traffic signals. This allowed us to expose subjects to a range of risky driving scenarios, including simulated car accidents, and track their reactions and perceptions of self-driving vehicles under different conditions. The contributions of this study are as follows: (1) We developed a realistic traffic VR system that provides an immersive self-driving vehicle experience with synchronized physical movement to the users. The developed system enables risky simulations which are impossible to be conducted in the real world, and enables complete control of road conditions to test the same scenario repeatedly. (2) We observed the UX change as the level of the information provided and the risk changed using the self-driving VR system. (3) We observed that the physiological and self-reported annotation data significantly change as the level of the information provided and the risk changes using the self-driving VR system. Eight scenarios were developed to observe the effects of two factors(level of risk, and level of information provision). We conducted experiments on 52 subjects and analyzed the quantitative and qualitative UX results to provide insights into designing self-driving vehicle experiences.

To assess trust in self-driving vehicles under these conditions, we considered both quantitative and qualitative metrics. We collected self-reported questionnaire data and galvanic skin responses (GSR). By comparing quantitative and qualitative data, we ensured the use of self-reported measures of valence and arousal. Valence and arousal(VA) are two commonly used dimensions of human emotion, and we validated whether our self-driving simulation provides an appropriate sense of emotion to users by observing VA annotation values. By analyzing the impact of information provision through HUDs at various risk levels in the VR simulation, we collected insights into how these factors influence the adoption of self-driving vehicles in the real world.

## 2. Related Works

### 2.1. VR Use Cases and Effects in Experiments Related to Autonomous Vehicles (AVs)

Virtual reality (VR) enables a simulated experience that allows users to engage as if they are in a real environment [11]. Based on this feature, various researches have conducted experiments using VR simulators, including observing pedestrian behavior [12], safety education [13], advancement of AV safety [14], and communication between AV and its surroundings [15]. By using VR for AV research, subjects can proceed in a realistic environment without exposure to the physical risks in the real environment. Thus, user behavior observation can be performed in completely controlled scenarios using VR simulations [14].

Deb et al. evaluated the suggestions made by pedestrians for the external features of fully autonomous vehicles (AVs) [12]. The study assessed the potential external features of a self-driving vehicle and identified the ones that would aid pedestrians in comprehending the actions of the vehicle at a crosswalk, boost their acceptance of self-driving cars, and affect how they cross the street. Nascimento et al. investigated the training and assessment of driving algorithms in VR environments [14]. The study aimed to comprehend the use of VR in boosting the safety of automated vehicles. Colley et al. has seven publications and preprints that discuss the use of VR in research on AV external communications [15]. Through these studies, it can be confirmed that VR is being used in fields which require controlled scenarios such as self-driving vehicle studies.

## 2.2. Prior Research on the Degree of Risk and the Reliability of Users According to Information Provision

Previous studies have investigated how risks affect users' trust before and after interaction with AVs. Experiments were conducted under four critical conditions (Very High Risk, High Risk, Low Risk, and No Risk) based on vehicle safety and integrity levels [9].

Morra et al. studied the virtual reality-driven simulators' role in trust building and human–machine interface (HMI) design for AVs [10]. Their research proposed a method to confirm the user experience in autonomous vehicles (AVs) using continuous and objective data obtained from physiological signals of the users during their immersion in VR driving simulations, demonstrating the usefulness of HMI design, specifically in the context of HUDs for AVs.

## 2.3. Quantitative Measurement of Valence and Arousal Using Galvanic Skin Response (GSR)

In emotion studies, valence and arousal are the two dimensions commonly used to represent human emotions. Valence refers to the degree of pleasantness or unpleasantness in the given emotion. In this study, an emotion with a higher valence value was considered to be positive. Arousal refers to the level of psychological activation. A higher arousal value indicates an intense feeling. Using the valence and arousal model, human emotions can be plotted on a two-dimensional plane, making it easier to assess emotions quantitatively.

Physiological signals were used to measure the valence and arousal values of the subjects. Typically, GSR measurements are used to infer changes in the emotional state of an individual because these changes can lead to change in sweat gland activity, which is reflected in the GSR signal. Several emotion studies have utilized GSR signals for automatically detecting valence and arousal. Tarnowski et al. invoked emotions by showing videos and collecting EEG and GSR signal data labeled by valence and arousal, and the GSR amplitude was correlated with valence and arousal values [16]. Raheel et al. used a machine learning approach to show the correlation between various physiological signals, such as EEG, GSR, PPG, valence, and arousal [17].

## 3. Study Design

### 3.1. Experimental Environment and Procedure

#### 3.1.1. AV Simulator

For immersive AV simulation, we used a virtual environment based on the Unity game engine and Oculus Quest 2 as the VR head-mounted display(HMD) as shown in Figure 1b. The HMD offers $1832 \times 1920$ resolution per eye, with 113.46 degrees of Field of View (FoV) and a display refresh rate of 120 Hz. The participants were asked to hold a VR controller in their hand, which is a joystick to annotate their valence and arousal. The functionality of the VR controller is discussed in Section 3.2. To create a motion simulator that reflected the size and movement of a real vehicle, a vehicle mock-up was manufactured and fixed to a PS-3TM-LP550 motion simulator [18]. The motion platform is shown in Figure 1a. This product can simulate three degrees of freedom (heave, roll, and pitch). The payload was 550 kg, the operating range was up to 0.14 m for the heave, the roll value was from −10.8 degrees to 10.8 degrees, and the pitch range was from −12.1 degrees to 13.1 degrees. The speeds were 0.276 m/s, 18 degree/s, and 22 degree/s for heave, roll, and pitch, respectively. The corresponding accelerations were 0.4 g, 250 degrees/s$^2$, and 250 degrees/s$^2$, for heave, roll, and pitch, respectively. An aluminum profile-based mock-up with a width of 1800 mm and a depth of 2500 mm was installed on the motion platform to produce an autonomous driving simulator almost identical in size to a small car. The movements of the vehicle in the virtual environment and the real simulator were matched by obtaining the angular velocity value from the vehicle model in the Unity engine and simulating it.

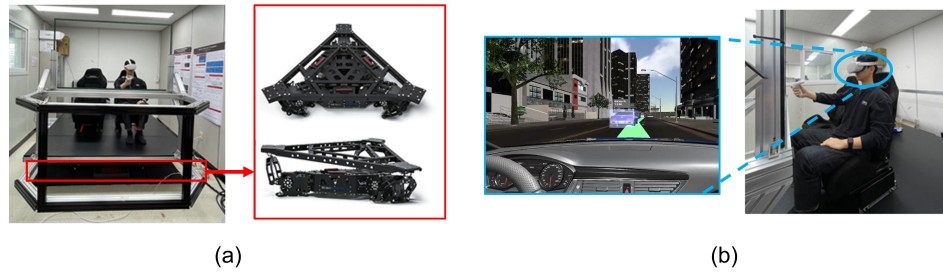

<div align="center">(a)        (b)</div>

**Figure 1.** (**a**) Front view of the Motion Platform. An aluminum profile-based mock-up was built to give realistic riding experience to the users. PS-3TM-LP550 motion was used to provide realistic motions of the virtual self-driving vehicle in the VR simulation. (**b**) Subject on Motion Platform wearing Oculus Quest 2 VR HMD, controlling the self-annotation panel with the joystick. The HMD offers a realistic view of the graphical simulations to the users.

### 3.1.2. Scenario

We provided eight scenarios according to the risk level (No Risk, Low Risk, Medium Risk, and High Risk) and whether the information was provided (Information Given, Information Not Given) to the user. Each scenario was approximately 1 min long, and an event occurred 10 s before the end of the scenario. All subjects were presented with the same simulated scenario. All the scenarios and the detailed description of each scenario is shown in Figure 2.

- No Risk—The self-driving vehicle drives normally from the beginning to the end of a route without being in any danger.
- Low Risk—The self-driving vehicle is not directly impacted. There is a sudden change in speed due to situations such as a sudden stop by an object appearing on the road.
- Medium Risk—The self-driving vehicle receives a direct weak impact. A minor accident is caused by another vehicle driving on the road.
- High Risk—The self-driving vehicle is directly impacted. A serious accident is caused by another vehicle driving on the road. A detailed description of the presence or absence of information provision of the AR display is as follows:
- Information Given—A state in which the self-driving vehicle displays the information and the route of the currently detected object on the HUD;
- Information Not Given—A state in which the AV provides no information to the HUD;

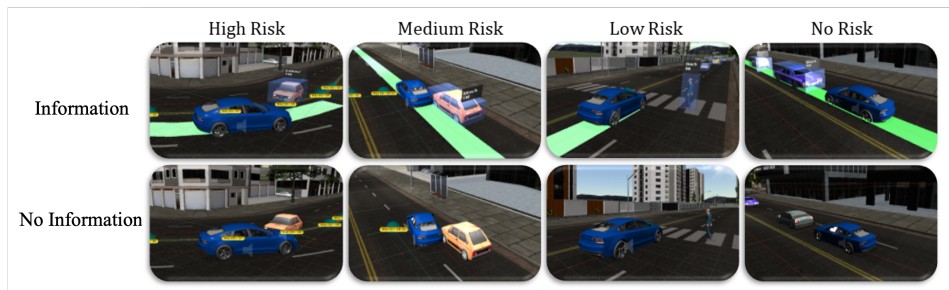

**Figure 2.** Eight scenarios categorized by level of risk and information provision on HUD.

### 3.1.3. Procedure

Fifty-two subjects with an average age of 21.2 and a standard deviation of 3.49 ($M$ = 21.2, $SD$ = 3.49) were recruited for this study and they received about 15 dollars after the experiments. Among 52 subjects, 25 were male and 27 were female. Using the Latin Square design, we randomized the eight scenarios in different orders for each subject in the experiment. Before boarding the motion platform, the subjects wore the Empatica E4 sensor and listened to a brief explanation of the progress of the experiment. GSR data were recorded at a sampling rate of 4 Hz through the E4 Realtime API in each scenario. After boarding the motion platform, each subject wore a VR HMD and was provided with

a VR controller. Subjects were asked to self-evaluate their emotions for the experience of the autonomous vehicle simulation with the VR controller along the two axes: Valence and Arousal. During the practice scenario, the subjects sufficiently practiced the emotion annotation and then proceeded to the main experiment. In cases where they experienced severe motion sickness, the experiments were terminated. The subjects were asked to complete an interim survey after each scenario; and a post-survey after all scenarios. The experimental timeline is shown in Figure 3.

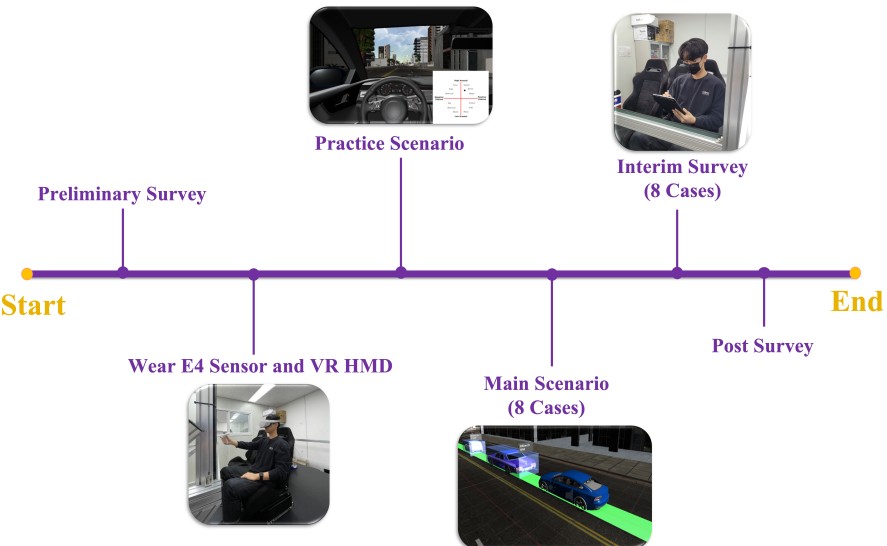

**Figure 3.** Timeline of the Experiment.

### 3.2. Data Collection and Statistical Analysis

We collected quantitative GSR data, self-reported valence and arousal annotation data, and qualitative data from a questionnaire. The subjects were asked to annotate valence and arousal in using the joystick of the VR controller. The subjects are able to move the black dot shown in Figure 4 (left) with their VR controller. In the front view inside the VR simulation, a panel consisting of two axes, valence, and arousal, is shown to the subject as Figure 4 (right). By moving the black dot with the VR controller, subjects are able to inform about their perceived emotions in real time. To help the subjects when performing the annotation, we provided labels of emotions in natural language (e.g., Excited, Happy, Sad, etc.), so the subjects were able to use these cues to directly perceive their feelings without having to think about valence and arousal.

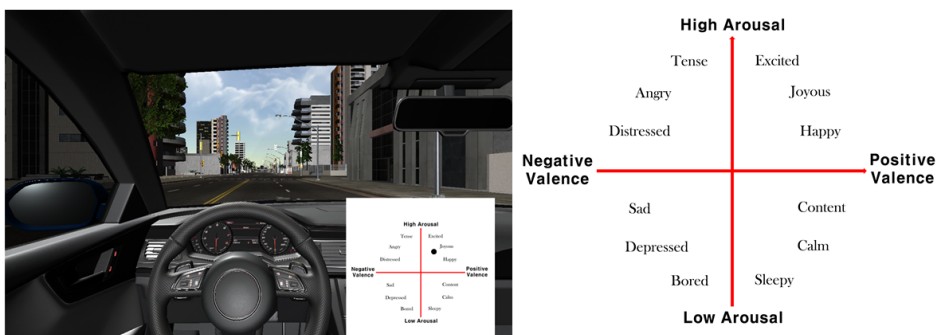

**Figure 4.** The front view of the subject and Valence and Arousal Annotation Graph.

### 3.2.1. Quantitative Measurement

Physiological data provide information on the user's emotional state, in particular, GSR data are sensitive to arousal and are important indicators of the user's arousal state. GSR data can be separated into skin conduction level (SCL) and skin conductance response

(SCR). SCL provides an overall emotional state, and SCR provides information on responses to specific stimuli [19,20]. The magnitude of SCR is related to a perceived threat [10]. Therefore, SCR was used to observe the responses to events. The MATLAB-based software Ledalab was used to extract the SCL and SCR features from the GSR raw data. Raw GSR data were pre-processed through an 8 Hz Gaussian Window and adaptive smoothing, and continuous decomposition analysis (CDA) was conducted to separate the GSR data into tonic (SCL) and phasic (SCR) elements. We also applied the min-max normalization to the GSR phasic data to minimize the differences between people. For each participant, the maximum and minimum values of GSR phasic data during the entire 8 scenarios were extracted and normalized to a value between 0 and 1 using the following Equation (1).

$$NormalizedX = \frac{X - Xmin}{Xmax - Xmin} \tag{1}$$

The following GSR data features and self-reported annotation data features were extracted before and after an event to isolate the data in response to the event. Data analysis was performed on 45 out of the 52 subjects. Four subjects with abnormal GSR data were excluded and three subjects were excluded from the analysis because the annotation data after the event occurred was missing.

- The maximum value of GSR data in the 10 s before and after the event (GSR Max)
- The average value of GSR data in the 10 s before and after the event (GSR Mean)
- The maximum value of GSR Tonic data in the 10 s before and after the event (GSR Tonic Max)
- The average value of GSR Tonic data in the 10 s before and after the event (GSR Tonic Mean)
- The maximum value of GSR Phasic data in the 10 s before and after the event (GSR Phasic Max)
- The average value of GSR Phasic data in the 10 s before and after the event (GSR Phasic'Mean)
- The maximum value of arousal data in the 10 s before and after the event (Arousal Max)
- The average value of arousal data in the 10 s before and after the event (Arousal Mean)
- The minimum value of arousal data in the 10 s before and after the event (Arousal Min)
- The maximum value of valence data in the 10 s before and after the event (Valence Max)
- The average value of arousal data in the 10 s before and after the event (Valence Mean)
- The minimum value of valence data in the 10 s before and after the event (Valence Min)

### 3.2.2. Qualitative Measurement

We surveyed user experiences of each scenario in autonomous driving simulators. We measured trust, perceived safety, immersion and presence, situational awareness, and reaction to events using questionnaires to evaluate the scenario experience. A list of questions is presented in Tables 1–4.

After the subjects experienced all eight scenarios, we investigated whether there was a difference in experience according to the level of risk and information provision on HUD (Table 5).

**Table 1.** Trust [21] and Perceived Safety [22]: Questionnaire list.

| Component | Item | Questionnaire |
|---|---|---|
| Trust | 1 | It may be on an autonomous vehicle |
| | 2 | Autonomous vehicles are reliable |
| | 3 | Overall, I trust autonomous vehicles. |
| Perceived Safety | 4 | I felt it would be dangerous to use an autonomous vehicle |
| | 5 | I felt safe while using the vehicle |
| | 6 | I believe in this vehicle |

**Table 2.** Immersion and Presence [23]: Questionnaire list.

| Item | Questionnaire |
|---|---|
| 1 | I felt a sense of being immersed in the virtual environment |
| 2 | I did not need to feel immersed in the virtual environment to complete my task |
| 3 | I had a sense of presence (i.e., being there) |
| 4 | The quality of the image reduced my feeling of presence |
| 5 | I thought that the field of view enhanced my sense of presence |
| 6 | The display resolution reduced my sense of immersion |
| 7 | I felt isolated and not part of the virtual environment |
| 8 | I had a good sense of scale in the virtual environment |
| 9 | I often did not know where I was in the virtual environment |
| 10 | Overall I would rate my sense of presence as: very satisfactory, satisfactory, neutral, unsatisfactory or very unsatisfactory |

**Table 3.** Situation awareness [24]: Questionnaire list.

| Item | Questionnaire |
|---|---|
| 1 | How changeable is the situation? |
| 2 | How complicated is the situation? |
| 3 | How many variables are changing within the situation? |
| 4 | How aroused are you in the situation? |
| 5 | How much are you concentrating on the situation? |
| 6 | How much is your attention divided in the situation? |
| 7 | How much mental capacity do you have to spare in the situation? |
| 8 | How much information have you gained about the situation? |
| 9 | How familiar are you with the situation? |

**Table 4.** Reaction to Events [10]: Questionnaire list.

| Item | Questionnaire |
|---|---|
| 1 | The situation was dangerous |
| 2 | The event took me by surprise |
| 3 | I was able to perceive the potential danger before it affected the vehicle's performance |
| 4 | The interface provided me useful information to foresee the event |

**Table 5.** Post Questionnaire list.

| Item | Questionnaire |
|---|---|
| 1 | In a driving situation without any risk (a scenario in which no accident occurred), a vehicle in which information was provided on HUD was felt to be safer. |
| 2 | In a low-risk driving situation (sudden stop by a pedestrian), a vehicle in which information was provided on HUD was felt to be safer. |
| 3 | In a medium-risk driving situation (direct weak impact by a car coming from behind), a vehicle in which information was provided on HUD was felt to be safer. |
| 4 | In a high-risk driving situation (serious car accident), a vehicle in which information was provided on HUD was felt to be safer. |
| 5 | In the presence of AR information, the trust in autonomous vehicle increase as the risk level of the scenario increased. |
| 6 | In the absence of AR information, the trust in autonomous vehicle increase as the risk level of the scenario increased. |

### 3.2.3. Statistical Analysis

Repeated analysis of variance (ANOVA) was conducted using JASP to confirm the significance of the quantitative and qualitative evaluations for all eight scenarios. Post-hoc comparisons were performed by applying Bonferroni correction. When the *p*-value was 0.05 or less, a statistically significant difference was determined. Python was used for GSR and VA data analysis and feature extraction.

## 4. Results

### 4.1. GSR Analysis Data

The features from the GSR data increased as the risk level of the scenario increased, indicating that the risk scenarios that we implemented were designed properly to invoke the intended relative level of intense emotions. Thus, we were able to validate that the scenarios designed with higher risk induced intense feelings in the subjects. We analyzed the GSR phasic data in the 10 s before and after the event in all eight cases, and extracted the maximum and mean values . We performed repeated measures ANOVA analysis on these features for a total of eight scenarios using JASP software. The analysis revealed significant differences according to the risk level for the two features of GSR Phasic Max and GSR Phasic Mean as shown in Figure 5: GSR Phasic Max, $F(3, 132) = 34.871$, $p < 0.001$; GSR Phasic Mean, $F(3, 132) = 25.567$, $p < 0.001$. We conducted the post-hoc analysis to find the change in GSR feature values according to the level of risk and information on the HUD. In the case of GSR Phasic Max, the high-risk condition was significantly higher than the medium-risk ($p = 0.003$), low-risk ($p < 0.001$), and no-risk conditions ($p < 0.001$). In the case of medium risk, the low-risk ($p < 0.001$) and no-risk conditions were significantly higher than that in the control condition ($p < 0.001$). In addition, when the AR information was provided, the value of GSR Phasic Max was slightly higher than that when the information was not provided ($p = 0.350$). In the case of GSR Phasic Mean, the high-risk condition was significantly higher than the medium-risk ($p = 0.001$), low-risk ($p < 0.001$), and no-risk conditions ($p < 0.001$). The medium-risk condition was significantly higher than the low-risk ($p < 0.001$) and the no-risk ($p = 0.004$) conditions. In addition, when information was provided, the value of GSR Phasic Mean was slightly higher than when the information was not provided ($p = 0.735$). From these results, we can validate the design of the risk scenarios.

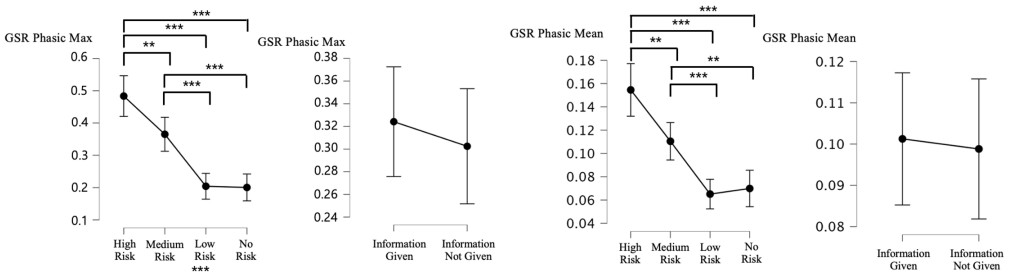

**Figure 5.** A graph that compares the maximum, mean, and accumulation value of GSR phasic data according to risk level and whether or not information was provisioned through HUD. The use of asterisks in the figure indicates statistical significance. Two (**) denotes a *p*-value less than 0.01, while three (***) asterisks denote *p*-values less than 0.001, respectively.

We investigated any significant changes in the GSR phasic data before and after the event. Figure 6 left shows the average GSR phasic data in the 10 s before and after the event for each case for the 45 subjects. From the graph, we can see that there is a rapid change in the GSR phasic value due to the event occurrence for high-risky and medium-risky conditions. In order to compare statistical values, we compared the maximum values of GSR phasic data, 10 s before and after the event occurred. We also found that there was a significant increase in the GSR phasic value in the 10 s before the event for 6 cases (Figure 6 right). A *t*-test was conducted to confirm the significant relationship between data before and after the event: high-risk and no-information, $t(44) = -6.754$, $p < 0.001$; high-risk and information, $t(44) = -7.481$, $p < 0.001$; medium-risk and no-information, $t(44) = -6.006$, $p < 0.001$; medium-risk and information, $t(44) = -6.681$, $p < 0.001$; low-risk and no-information, $t(44) = -2.195$, $p = 0.033$; low-risk and information, $t(44) = -2.521$, $p = 0.015$. In other words, the biometric data showed a rapid change according to our risk level, which implies that autonomous vehicle research using our motion simulation system is quite meaningful.

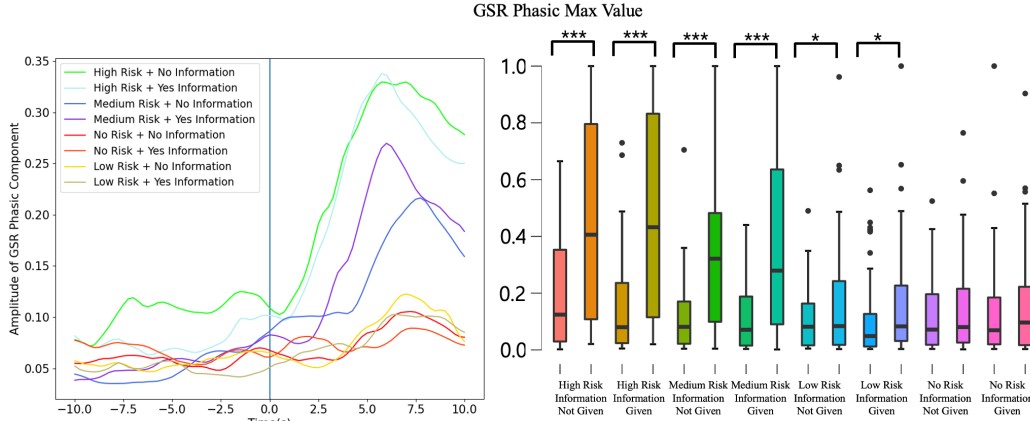

**Figure 6.** Left: Average values across all subjects for GSR phasic data from 10 s before and after the event for the eight scenarios. Right: A graph comparing GSR phasic data before and after the event. The use of asterisks in the figure indicates statistical significance. A single asterisk (*) denotes a *p*-value less than 0.05, while three (***) asterisks denote *p*-values less than 0.001, respectively.

### 4.2. Self-Reported Valence and Arousal Data

Through the self-reported VA data, we found that there were significant changes in participants' emotions according to the risk level and information provision level. We analyzed the self-reported VA data in the 10 s before and after the event in eight cases. The maximum, minimum, and mean values of the VA data were obtained, and analysis was conducted through repeated measures ANOVA. Figures 7 and 8 show that, according to the risk level, significant differences were observed for Arousal Max, Arousal Mean, Valence Min, and Valence Mean: Max Arousal, $F(3, 132) = 41.043$, $p < 0.001$; Mean Arousal, $F(3, 132) = 20.057$, $p < 0.001$; Min Valence, $F(3, 132) = 29.555$, $p < 0.001$; Mean Valence, $F(3, 132) = 14.482$, $p < 0.001$. A post-hoc analysis was conducted on the risk level and whether or not the information was provisioned through HUD. In the case of Max Arousal, the high-risk condition was significantly higher than the medium-risk condition ($p < 0.001$), low-risk condition ($p < 0.001$), and no-risk condition ($p < 0.001$), and the medium-risk condition was significantly higher than the no-risk condition ($p < 0.001$). In addition, when information was provided to HUD, the value of Arousal Max was significantly lower than when the information was not provided ($p = 0.004$). In the case of Mean Arousal, the high-risk condition was significantly higher than the low-risk ($p < 0.001$), and no-risk conditions ($p < 0.001$); and the medium-risk condition was significantly higher than the low-risk condition ($p < 0.001$) and the no-risk condition ($p < 0.001$). In addition, when information was provided, the value of Mean Arousal was slightly lower than when the information was not provided ($p = 0.018$). In the case of valence min, the high-risk condition was significantly lower than the medium-risk condition ($p < 0.001$), low-risk condition ($p < 0.001$), and no-risk condition ($p < 0.001$); and the medium-risk condition was significantly lower than the no-risk condition ($p < 0.001$); and the low-risk condition was significantly lower than the no-risk condition ($p < 0.001$). In addition, when information was provided, the value of Min Valence was significantly higher than when information was not provided ($p = 0.031$). In the case of Mean Valence, the high-risk condition was significantly lower than the low-risk condition ($p < 0.001$) and the no-risk condition ($p < 0.001$), and the medium-risk condition was significantly lower than the no-risk condition ($p = 0.001$). When information was provided, the value of the Mean Valence was slightly higher than when the information was not provided ($p = 0.074$). As we intended from our scenario design, the valence score decreased and the arousal score increased as the risk level increased. This indicates that the subjects had negative emotions that were physiologically activated as the intensity of risk increased. In the case of high-risk conditions, the maximum value of arousal was significantly higher and the minimum value of valence was significantly lower than in medium-risk, low-risk, and no-risk conditions. In other words, we could find that the subjects actually had a sudden

change in emotion for the accident simulation situation. Moreover, there was no significant difference in VA mean value between medium-risk and low-risk conditions, but there was a significant difference in VA mean value between low-risk and no-risk conditions. In other words, we can simulate the different levels of risk through our motion simulation system. We also found that the level of information provision affected the emotion of participants. When the information was provided to the HUD, the participants showed low arousal and high valence. In other words, it was found that providing information to participants in an autonomous vehicle had a positive effect on participants' emotions.

We also observed significant changes in VA mean values before and after the event. A t-test was conducted to confirm the significant relationship between data before and after the event. In the case of Arousal Mean, the value increased significantly after the event occurred than before the event occurred in 7 cases: high-risk and no-information, $t(44) = -8.786$, $p < 0.001$; high-risk and information, $t(44) = -8.238$, $p < 0.001$; medium-risk and no-information, $t(44) = -7.008$, $p < 0.001$; medium-risk and information, $t(44) = -7.561$, $p < 0.001$; low-risk and information, $t(44) = -5.740$, $p < 0.001$; low-risk and no-information, $t(44) = -6.738$, $p < 0.001$; no-risk and information, $t(44) = -4.073$, $p < 0.001$; In the case of Valence Mean, the values after the event significantly decreased compared to that before the event occurred in 7 cases: high-risk and no-information, $t(44) = 5.175$, $p < 0.001$; high-risk and information, $t(44) = 4.462$, $p < 0.001$; medium-risk and no-information, $t(44) = 4.430$, $p < 0.001$; medium-risk and information, $t(44) = 5.045$, $p < 0.001$; low-risk and information, $t(44) = 3.137$, $p = 0.003$; low-risk and no-information, $t(44) = 3.078$, $p < 0.004$; no-risk and information, $t(44) = 2.706$, $p = 0.010$; Through the above statistical analysis, it was possible to decipher changes in the participants' emotions according to the risk level through simulation, and we observed that there were actually significant changes in the participants' annotation data and biometric data before and after events.

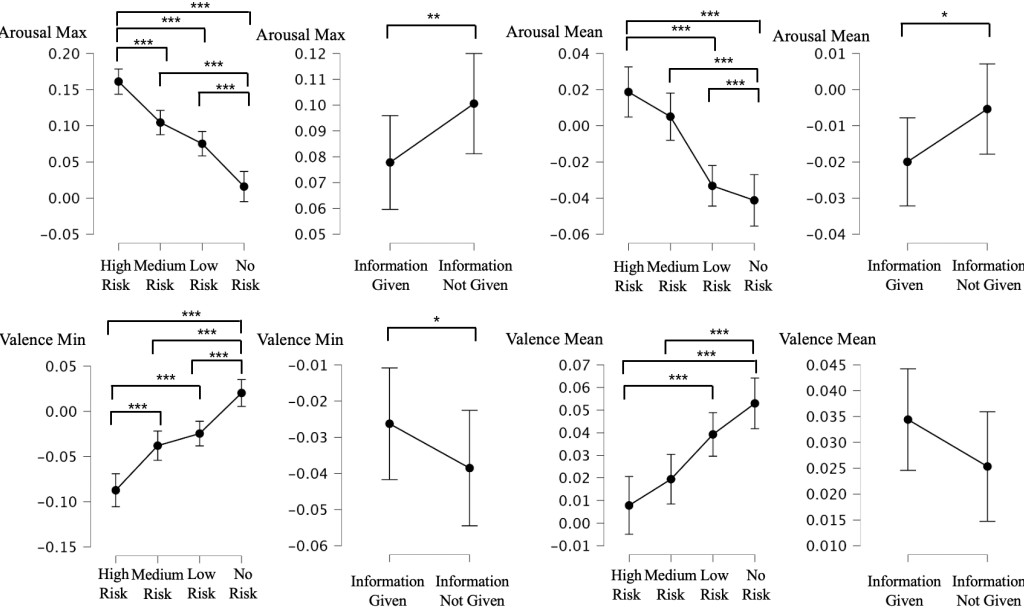

**Figure 7.** A graph that compares the maximum and mean value of Arousal data and the minimum and mean value of Valence data according to risk level and whether or not information was provisioned through HUD. The use of asterisks in the figure indicates statistical significance. A single asterisk (*) denotes a $p$-value less than 0.05, while two (**) and three (***) asterisks denote $p$-values less than 0.01 and 0.001, respectively.

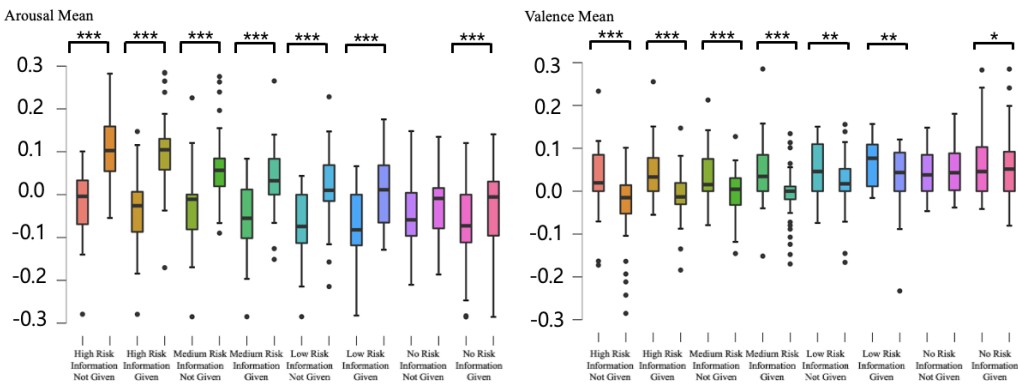

**Figure 8.** A graph comparing Arousal mean and Valence mean data before and after the event. The use of asterisks in the figure indicates statistical significance. A single asterisk (*) denotes a *p*-value less than 0.05, while two (**) and three (***) asterisks denote *p*-values less than 0.01 and 0.001, respectively.

### 4.3. Analysis of the Questionnaire Data

Questionnaires on the user experience were completed after each of the eight scenarios. For the questionnaire results on trust, perceived safety, presence, situation awareness, and reaction to events, the significance according to risk and information provision was confirmed through repeated measures ANOVA. The results are presented in Figures 9 and 10. There was a significant difference for trust, perceived safety, situation awareness, and reaction according to the degree of risk: trust, $F(3, 132) = 26.863$, $p < 0.001$; perceived safety, $F(3, 132) = 13.660$, $p < 0.001$; situation awareness, $F(3, 132) = 45.982$, $p < 0.001$; reaction question 1, $F(3, 132) = 141.740$, $p < 0.001$; reaction question 2, $F(3, 132) = 92.155$, $p < 0.001$; reaction question 3, $F(3, 132) = 20.645$, $p < 0.001$; and reaction question 4, $F(3, 132) = 11.176$. In addition, there was a significant difference in trust, situation awareness, and reaction depending on whether the information was provided: trust, $F(1, 44) = 7.278$, $p < 0.05$; situation awareness, $F(1, 44) = 11.272$, $p < 0.01$; reaction question 3, $F(1, 44) = 26.752$, $p < 0.001$; and reaction question 4, $F(1, 44) = 54.092$, $p < 0.001$.

Post-hoc analysis was conducted on changes in trust, perceived safety, situational awareness, and reaction to events according to the degree of risk and whether the information was provided. In the case of a trust, the high-risk condition was significantly lower than the low-risk ($p < 0.001$), and no-risk conditions ($p < 0.001$). In addition, when information was provided, the value of trust was significantly higher than when it was not provided please add explanation for the asteriks symbols ($p < 0.01$). In the case of perceived safety, the high-risk condition was significantly lower than the low-risk ($p < 0.001$) and no-risk conditions ($p < 0.001$). Moreover, when information was provided, the value of perceived safety was slightly higher than when information was not provided ($p = 0.109$). In case of the Immersion and Presence Questionnaire, there were no significant differences among the eight cases. However, data show that users had a high sense of immersion and presence of 4.0 or higher. In the case of situation awareness, the high-risk condition was significantly lower than the medium-risk ($p < 0.05$), the low-risk please add explanation for the asteriks symbols ($p < 0.001$), and no-risk conditions ($p < 0.001$), and the medium-risk condition was significantly lower than the low-risk condition ($p < 0.001$) and no-risk condition ($p < 0.001$). In addition, when the information was provided, the situation awareness was significantly higher than when the information was not provided ($p < 0.001$). Through the survey results, the GSR biometric data, and the self-reported VA annotation data, it was found that the trust, perceived safety, and situation awareness felt by the subjects changed significantly according to the risk level. In addition, since the results of the Immersion and Presence questionnaire were high scores of almost 4.0 on average, we can infer that the subjects felt a sense of immersion and presence enough for the simulation situation. In addition, when

the information was provided through HUD, the value of trust and situation awareness was significantly higher than when the information was not provided, and it was found that providing information from autonomous vehicles has a positive effect on trust and situation awareness.

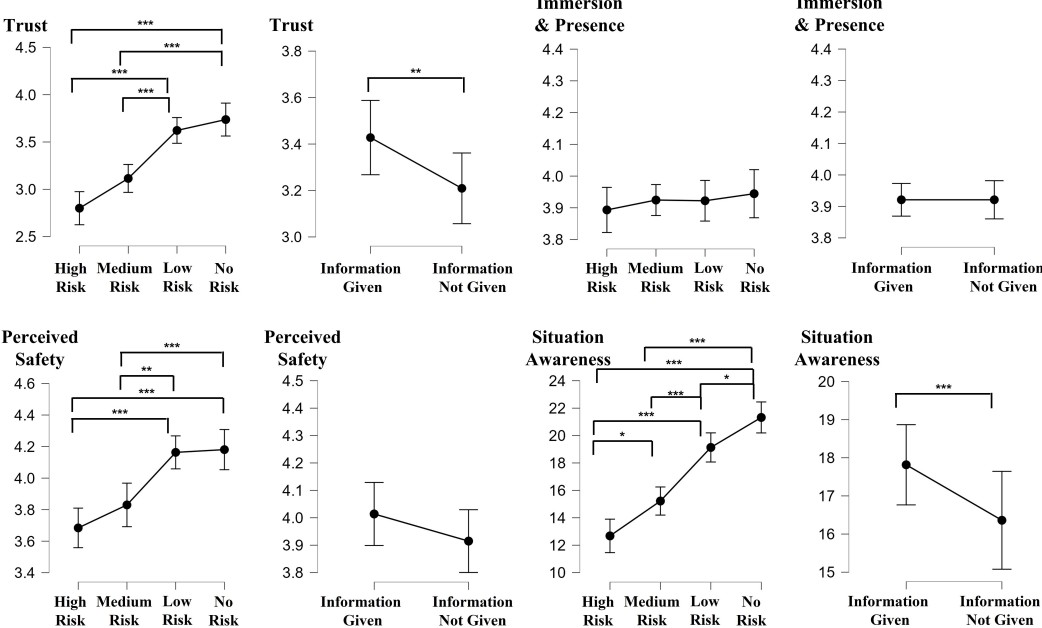

**Figure 9.** A graph that compares trust, perceived safety, immersion and presence, and situation awareness questionnaire data according to risk level and whether or not information was provisioned through HUD. The use of asterisks in the figure indicates statistical significance. A single asterisk (*) denotes a *p*-value less than 0.05, while two (**) and three (***) asterisks denote *p*-values less than 0.01 and 0.001, respectively.

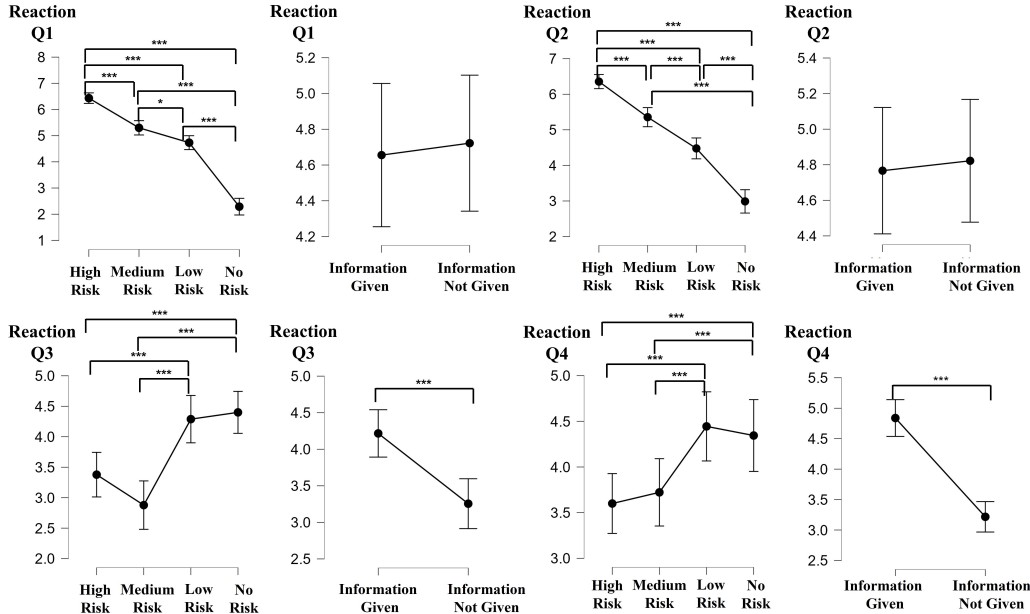

**Figure 10.** A graph that compares reaction to events questionnaire data according to risk level and whether or not information was provisioned through HUD. The use of asterisks in the figure indicates statistical significance. A single asterisk (*) denotes a *p*-value less than 0.05, while three (***) asterisks denote *p*-values less than 0.001, respectively.

Through the reaction to the events questionnaire, we found that there was a significant difference in the degree of risk perceived by the subjects according to the risk level and that the degree of awareness of the upcoming risk significantly increased when information was provided to HUD. In the case of reaction questionnaire 1, the high-risk condition was significantly higher than the medium-risk ($p < 0.001$), low-risk ($p < 0.001$), and no-risk ($p < 0.001$) conditions and the medium-risk condition was significantly higher than the low-risk ($p < 0.05$) and no-risk ($p < 0.001$) conditions and the low-risk condition was significantly higher than the no-risk condition ($p < 0.001$). In the case of reaction questionnaire 2, the high-risk condition was significantly higher than the medium-risk condition ($p < 0.001$), low-risk condition ($p < 0.001$), and no-risk condition ($p < 0.001$), and the medium-risk condition was significantly higher than the low-risk condition ($p < 0.001$) and the no-risk condition ($p < 0.001$), and the low-risk condition was significantly higher than the no-risk condition ($p < 0.001$). In the case of reaction questionnaire 3, the high-risk condition was significantly lower than the low-risk ($p < 0.001$) and no-risk conditions ($p < 0.01$), and the medium-risk condition was significantly higher than the low-risk ($p < 0.001$) and no-risk conditions ($p < 0.001$). In addition, when information was provided, the value was significantly higher than when the information was not provided ($p < 0.001$). The score of reaction questionnaire 4 was significantly higher when the information was provided than when the information was not provided ($p < 0.001$).

For post-questionnaire 1–4, we confirmed the significance using ANOVA. Except for the significance between high-risk and no-risk conditions ($p < 0.05$), there was no significant difference between risk levels, but the value decreased slightly from question 1 to question 4 (Figure 11 Left): $F(3, 176) = 3.727$, ($p < 0.05$). For post-questionnaire 5–6, we confirmed the significance using the Wilcoxon signed-rank test. The value for question 5 was significantly higher than that for question 6 ($Z = 3.677$, $p < 0.001$) (Figure 11 Right). Through post questions 1–4, we found that the higher the risk, the more the subject felt safe about providing information to the HUD. In addition, through post questions 5–6, we found that as the risk level increased, information provided on the HUD increased the trust in autonomous vehicles, while not information provided on the HUD decreased the trust. In other words, we could infer that information provided on the HUD had a positive effect on the user experience of autonomous vehicles.

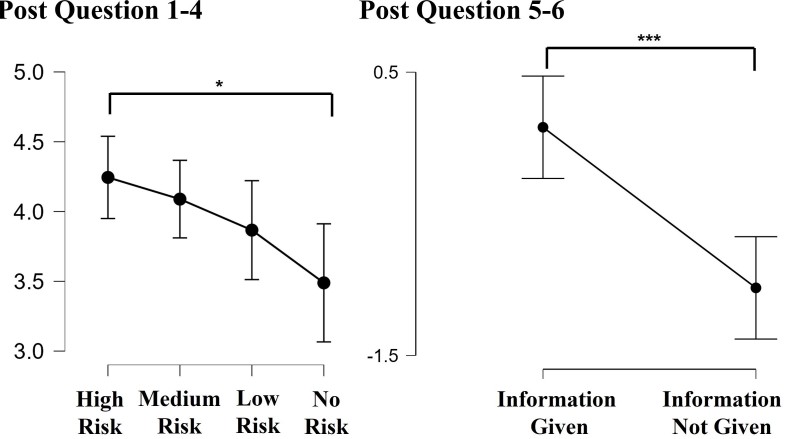

**Figure 11.** A graph that compares post-questionnaire data according to risk level and whether to provide information to HUD or not. The use of asterisks in the figure indicates statistical significance. A single asterisk (*) denotes a *p*-value less than 0.05, while three (***) asterisks denote *p*-values less than 0.001, respectively.

## 5. Discussion

### 5.1. Risk Simulation Design

Our study aimed to investigate the user experience with regard to different levels of risk and information provision. We utilized a VR simulation to control the road environment and simulate risky situations that are impossible to simulate in the real world. Because we designed the environment and scenario in the virtual simulation, the designed simulation was able to evoke the intended emotions in the users. To validate this factor, we asked the subjects to annotate their emotions with valence and arousal values while recording GSR signals using the Empatica E4 wristband. Physiological signals are commonly used to sense the current state of human subjects, such as their emotions, and some researchers have attempted to predict emotions from physiological sensor data [25].

We validated our simulation by comparing GSR and self-reported VA data. We analyzed GSR signals by extracting the GSR phasic features to quantitatively observe the emotions of the subjects. The GSR signals showed significant changes before and after the risk event, which ensured that the designed scenario caused a change in the emotions of the subjects. In addition, by analyzing the self-reported valence and arousal data, we found that, as the scenario's risk level increased, the overall valence score decreased, while the overall arousal score increased. From these observations, we can conclude that the subjects felt a strong negative feeling when the risk level was high than when it was low.

### 5.2. The Effect of Risk Level

Quantitative measurements show that according to the risk level, there are significant changes in the GSR physiological data and self-reported VA values of users. As reported in previous studies [10], it was found that the higher the risk level, the higher the GSR phasic value, and the subjects also reported that their level of arousal was high and the valence was low. Qualitative measurements show that the risk level is a critical factor in user experience. We measured user trust, perceived safety, and situational awareness using questionnaires after each scenario. The overall score of each user experience decreased as the risk level increased. The immersion and presence were almost equal at all risk levels, indicating that all scenarios were similarly realistic. Thus, there was no effect on the user experience owing to differences in the completeness of each VR scenario. The results of reaction questions 1 and 2 additionally validated our scenario designs. The users felt that the situation was more danerous and surprising, as we provided scenarios that were intended to be riskier. Through reaction questions 3 and 4, the users showed significantly higher awareness of the potential risk. This confirmed that providing road-related situational information to users continuously improved the perception of road conditions.

### 5.3. The Effect of Information Provision

To understand the user experience of the simulated self-driving system better, we conducted post-hoc surveys that inquired about the helpfulness of providing information directly to the users. Figure 11 shows the subjects' preferences for the cases where information was provided. There was a significant increase in perceived safety when the subjects were provided with information. Because transparent decision-making is considered important for building user trust in self-driving vehicles [26], this result shows that visualizing how the vehicle perceives the road conditions helps improve user experience.

The trust data results follow those of previous studies [10]. Overall, users exhibited significantly higher trust when information on the road situation was presented on the HUD. Our results, which were all counterbalanced , emphasize the importance of providing information since it improved trust in all situations and at every risk level. Because information provision resulted in increased trust in various levels of risk, explaining the system's perception of current road situations is a critical factor for building trust between the self-driving vehicle and the user. Also, the users showed significantly higher situational awareness when the information provision using HUDs. Since increased situational awareness can lead to an increase in the driver's trust towards self-driving vehicles [27],

this result emphasizes that providing information about road conditions is crucial for a better user experience . The self-reported Valence and Arousal data also indicate that the provision of information through HUD reduces the user's arousal and changes the valence to a positive one. In other words, providing real-time surrounding information about autonomous vehicles, such as traffic environments, can enhance the user experience.

## 6. Limitations and Future Works

### 6.1. Limitations

In this exploratory study, we leveraged an immersive virtual reality (VR) simulation to investigate user experiences in various high-risk scenarios. By analyzing physiological signals such as heart rate and galvanic skin response as well as self-reported questionnaires, we demonstrated that our carefully designed VR simulations were capable of evoking appropriate levels of stress and anxiety in participants as they encountered dangerous situations. However, the scope of scenarios in our study was limited. We implemented a single VR simulation for each combination of risk level and information provision, resulting in a total of eight VR simulations. While our scenarios were designed based on findings from previous studies [9,10], a more comprehensive investigation would necessitate a wider range of scenarios in order to fully capture the diverse array of accidents and high-risk situations that arise in the real world. A larger set of scenarios would allow for a robust analysis of psychological and physiological responses across different conditions as well as more nuanced insights into the design of VR systems for research and training purposes.

### 6.2. Future Works

VR simulation enables the repeated simulation of driving conditions while enabling intended emotions to be evoked in users. In addition, because VR simulation enables us to control every variable, it excludes the possible effects of extraneous variables and enables us to observe clearly the effects of the specified variables. For example in the follow-up study, additional features such as various accident scenarios, weather and temperature conditions, and various road scenes can be added in our study. The user study on diverse conditions of VR scenarios would guarantee more generalized insights into how user experience would change on various levels of information provision.

Recent studies have used neural networks in detecting user experience. These methods analyze data such as facial expressions [28–30], physiological signal [31–33], and content-wise affection analysis [34,35]. The neural network-based analysis directly infers human experience such as emotion, thus providing more insights toward understanding user experience. In this study, we utilized GSR signals to analyze the emotional change of the users while immersed in VR simulations. By introducing neural networks to our study, fine-grained user experience analysis is possible in addition to statistical analysis of the GSR signals.

## 7. Conclusions

In this study, we proposed a motion platform-based VR simulation method to provide driving situations with different levels of risk and information provision on HUD and evaluated the user experience based on GSR physiological data, self-reported VA data, and questionnaires on trust, perceived safety, situation awareness, immersion and presence, reaction to events, and custom question. The results show that user experience changes significantly according to the level of risk and information provision on HUD. We showed that UX evaluation is possible using VR simulations using our methodology. Through the analysis of GSR physiological data and self-reported valence and arousal annotation data, we were able to find out how participants actually responded to risk levels. Through analysis of questionnaire reports, we found that the trust and the situation awareness, which are crucial factors when accepting the self-driving vehicles, were significantly increased by presenting the information of road situations on the AR HUDs. Our result emphasizes that

explaining the perception of the self-driving vehicles towards the on road situations is a key factor for improving the user experience of the self-driving vehicles.

In future, we plan to conduct simulations including various situations such as diverse weather and temperature conditions, the number of vehicles on the road, and take-over request (TOR)related tasks. In addition, we plan to conduct research on situations in which multiple users can physically interact by taking advantage of our simulator, which can accommodate two or more people.

**Author Contributions:** Conceptualization, J.O.; methodology, S.K. (Seungju Kim) and M.S.; software, S.K. (Seungju Kim) and J.O.; formal analysis, M.S.; data curation, M.S.; writing—original draft preparation, S.K. (Seungju Kim), J.O. and M.S.; writing—review and editing, E.J.; visualization: M.S. and J.O.; supervision, S.K. (Seungjun Kim); funding acquisition, Y.-K.M. All authors have read and agreed to the published version of the manuscript.

**Funding:** This research was supported by Culture, Sports and Tourism R&D Program through the Korea Creative Content Agency grant funded by the Ministry of Culture, Sports and Tourism in 2022 (Project Name: Development of Interactive Contents and Platform based on Multiple Scenario in Autonomous Vehicle, Project Number: R2020040058, Contribution Rate: 100).

**Institutional Review Board Statement:** The study was conducted according to the guidelines of the Declaration of Helsinki, and approved by the Institutional Review Board of Gwangju Institute of Science and Technology. (Protocol code 20210609-HR-61-02-02, approved on 21 June 2021).

**Informed Consent Statement:** Informed consent was obtained from all subjects involved in the study.

**Data Availability Statement:** The data presented in this study are available on request from the corresponding author.

**Conflicts of Interest:** The authors declare that there is no conflict of interest.

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
