# Peer review of "Assessing the Impact of AR HUDs and Risk Level on User Experience in Self-Driving Cars: Results from a Realistic Driving Simulation"

_applsci, doi:10.3390/app13084952_

Round 1

Reviewer 1 Report

1.The manuscript is divided into different risk levels, but the relevant risk level scenarios provided by the manuscript are relatively single and are not diversified. It is suggested that the author should supplement this part.

2.The pictures shown in the manuscript are vague, such as Figure 5, etc. It is suggested that the author should replace or adjust them.

3.The number of subjects involved in the manuscript was small, and due to the contingency of the experiment, 32 subjects cannot guarantee the accuracy of the experimental results. It is recommended to increase the number of subjects to avoid the contingency of the experiment.

4.The authors should be rechecking the grammatical errors and typos issues in the complete manuscript.

Author Response

Response to Reviewer 1 Comments

Dear reviewer 1,

Thank you for your feedback. We have revised our paper in response to your comments.
Please find our response document attached. We have addressed each of your points and clarified or revised sections as needed.
We appreciate your constructive criticism, which has helped to improve our work.

Thank you.

Sincerely,

Seungju Kim, Jungseok Oh, Minwoo Seong, Eunki Jeon, and Yeon-kug Moon

Reviewer 2 Report

I feel the paper esponds to some questions on the safety of the self driving vehicles and is a background for further research with the use of e.g. neural networks. Great job.

Author Response

Response to Reviewer 2 Comments

Dear reviewer 2,

Thank you for your feedback. We have revised our paper in response to your comments.
Please find our response document attached. We have addressed each of your points and clarified or revised sections as needed.
We appreciate your constructive criticism, which has helped to improve our work.

Thank you.

Sincerely,

Seungju Kim, Jungseok Oh, Minwoo Seong, Eunki Jeon, and Yeon-kug Moon

Reviewer 3 Report

This paper proposed a motion platform-based VR simulation method to provide driving situations with different levels of risk and head-up display (HUD) information. User experience are evaluated with galvanic skin responses (GSR) physiological data, self-reported VA data, and questionnaires of trust. The results showed that user experience changes significantly according to the level of risk and information provided to HUD. The proposed VR simulation platform can simulate the autonomous vehicle to some extent. But the article also has obvious problems.

1. the relationship between the level of risk and the subjectss responses is obvious. The results given by the experiment are only demonstrative and meaningless.

2. because of the small number of subjects, the experiment lacks reliability. At the same time, the analysis of GSR and VA data in section 4.1, 4.2, and 4.3 did not give any valuable conclusions. It can only explain the rationality of the comparative experiment design, but cannot fully prove the effectiveness of the motion simulation system.

3. there are some small problems. The device held by the subject in Figure 1 is not introduced. Section 3.1.3 does not give the specific meaning of M and SD in (M=22.1, SD=4.30).

Author Response

Response to Reviewer 3 Comments

Dear reviewer 3,

Thank you for your feedback. We have revised our paper in response to your comments.
Please find our response document attached. We have addressed each of your points and clarified or revised sections as needed.
We appreciate your constructive criticism, which has helped to improve our work.

Thank you.

Sincerely,

Seungju Kim, Jungseok Oh, Minwoo Seong, Eunki Jeon, Yeon-kug Moon, and SeungJun Kim

Round 2

Reviewer 1 Report

I thank the authors for their updates. The paper has been considerably improved. However, I still recommend a MINOR REVISION. This is because the overall quality of the English text has improved but is not perfect. Moreover, some minor issues remain in the manuscript, as follows.

1. Some cited references are too old.

2. The manuscript lacks individual contributions by the six authors. It is suggested that the authors make additions.

Author Response

(The authors gave the same response as above.)

Reviewer 3 Report

I have no further comments.

Author Response

(The authors gave the same response as above.)
